# Profiles of Cytokinins Metabolic Genes and Endogenous Cytokinins Dynamics during Shoot Multiplication In Vitro of *Phalaenopsis*

**DOI:** 10.3390/ijms23073755

**Published:** 2022-03-29

**Authors:** Yuan-Yuan Li, Zhi-Gang Hao, Shuo Miao, Xiong Zhang, Jian-Qiang Li, Shun-Xing Guo, Yung-I Lee

**Affiliations:** 1Institute of Medicinal Plant Development, Chinese Academy of Medical Sciences & Peking Union Medical College, Beijing 100193, China; liyuanyuan184@163.com; 2College of Plant Protection/Beijing Key Laboratory of Seed Disease Testing and Control, China Agricultural University, Beijing 100193, China; 17810266056@163.com (Z.-G.H.); ms_0825@163.com (S.M.); lijq231@cau.edu.cn (J.-Q.L.); 3The Key Laboratory of Biology and Genetic Improvement of Oil Crops, The Ministry of Agriculture and Rural Affairs of the PRC, Oil Crops Research Institute of the Chinese Academy of Agricultural Sciences, Wuhan 430062, China; hbzhangxiong@126.com; 4Department of Life Science, National Taiwan University, Taipei 10617, Taiwan

**Keywords:** *Phalaenopsis*, micropropagation, cytokinins, gene expression

## Abstract

Shoot multiplication induced by exogenous cytokinins (CKs) has been commonly used in *Phalaenopsis* micropropagation for commercial production. Despite this, mechanisms of CKs action on shoot multiplication remain unclear in *Phalaenopsis*. In this study, we first identified key CKs metabolic genes, including six isopentenyltransferase (*PaIPTs*), six cytokinin riboside 5′ monophosphate phosphoribohydrolase (*PaLOGs*), and six cytokinin dehydrogenase (*PaCKXs*), from the *Phalaenopsis* genome. Then, we investigated expression profiles of these CKs metabolic genes and endogenous CKs dynamics in shoot proliferation by thidiazuron (TDZ) treatments (an artificial plant growth regulator with strong cytokinin-like activity). Our data showed that these CKs metabolic genes have organ-specific expression patterns. The shoot proliferation in vitro was effectively promoted with increased TDZ concentrations. Following TDZ treatments, the highly expressed CKs metabolic genes in micropropagated shoots were *PaIPT1*, *PaLOG2,* and *PaCKX4*. By 30 days of culture, TDZ treatments significantly induced CK-ribosides levels in micropropagated shoots, such as *t*ZR and iPR (2000-fold and 200-fold, respectively) as compared to the controls, whereas *c*ZR showed only a 10-fold increase. Overexpression of *PaIPT1* and *PaLOG2* by agroinfiltration assays resulted in increased CK-ribosides levels in tobacco leaves, while overexpression of *PaCKX4* resulted in decreased CK-ribosides levels. These findings suggest de novo biosynthesis of CKs induced by TDZ, primarily in elevation of *t*ZR and iPR levels. Our results provide a better understanding of CKs metabolism in *Phalaenopsis* micropropagation.

## 1. Introduction

*Phalaenopsis* is the leading commercial potted plants worldwide because of its beautiful and long-lasting flowers. *Phalaenopsis* breeders frequently produce new cultivars with novel traits using hybridization; as a result, the selected elites are usually heterozygous. In order to provide uniform *Phalaenopsis* plantlets for growers, plant tissue culture technology is critically important for mass clonal propagation in *Phalaenopsis* commercial production [1]. In *Phalaenopsis* micropropagation, the nodal segment form inflorescence is the most common source material for subsequent clonal propagation in vitro [2]. The multiplication rate can be affected by the basal medium compositions, the balance of plant growth regulators and genotypes, etc. [3].

Among the plant growth regulators used in tissue culture, cytokinins (CKs) are key regulators in cell division and differentiation in plants [4,5]. Naturally occurring CKs are free-base forms of zeatin (Z) and adenine derivatives, i.e., dihydrozeatin (DZ) and *N*^6^-(Δ^2^-isopentenyl) adenine (iP), and their ribosides (ZR, DZR, and iPR, respectively) [4,5]. Moreover, aromatic CKs, including benzyladenine (BA) and its derivatives and topolins, the meta-topolin (*m*T), ortho-topolin (*o*T), and para-topolin (*p*T) families, can be detected in various conjugated forms within the plants [4,5]. In plant tissue culture, within the explant, its CKs come from the endogenous biosynthesis and the exogenous supply in the medium. The uptake and metabolism of both exogenous and endogenous CKs determine the overall CKs pool within the explant, thereby affecting growth and development of the plant in vitro [6,7]. Moreover, the metabolism of endogenous CKs can be strongly influenced by exogenous CKs, such as BA, *m*T, or TDZ used in culture media [7,8,9]. In CKs metabolism, CKs may be reversibly or irreversibly conjugated with sugars and amino acids [4,10,11]. Often, the exogenous CKs are converted into various forms of metabolites as a regulatory mechanism to maintain CKs homeostasis [11,12].

The biosynthesis of CKs involves three key enzymes: isopentenyltransferase (IPT), cytochrome P450 enzyme CYP735A (CYP735A), and LONELY GUY (LOG). These enzymes act in sequence to add a prenyl group to the *N*^6^ position of ADP/ATP, hydroxylate the isoprenoid side chain, and then activate the CK by converting it to the free-base form [12,13]. IPT is the rate-limiting enzyme involved in CK biosynthesis [14]. There are two classes of IPTs, in which ATP/ADP IPTs are responsible for the majority of *trans*-zeatin type (*t*Z type) and isopentenyladenine-type (iP type) of CK synthesis, whereas tRNA IPTs are required for *cis*-zeatin type (*c*Z type) CK production by degradation of cis-hydroxy isopentenyl tRNAs [5,15]. On the contrary, cytokinin oxidase (CKX) catalyzes irreversible inactivation of CKs, lowering their cellular level. These CKs metabolic enzymes are encoded by small multigene families, such as IPTs, LOGs, and CKXs. In *Arabidopsis*, spatiotemporal expression analyses of these metabolic enzymes demonstrated tissue- and organ-specific distribution patterns [16].

The key CKs metabolic genes have been identified in many plants, including *Arabidopsis*, *Oryza sativa*, *Zea mays*, soybean, and tomato [14,17,18,19,20,21,22], but not in *Phalaenopsis*. *Phalaenopsis aphrodite* is an important species in breeding programs that has a relatively small genome size (1.2 Gb) with validated chromosome-scale orchid genome assembly data [23], which greatly facilitates the analysis of gene expression profiles in *Phalaenopsis*. A better understanding of their expression profile and responses to exogenous plant growth regulators during tissue culture may help to unravel the molecular mechanisms involved in hormonal control in *Phalaenopsis* micropropagation.

In past few decades, much progress has been made in tissue culture protocols for *Phalaenopsis* micropropagation. Many reports have focused on the types and optimization of plant growth regulator conditions in accelerating multiplication [1]. Thidiazuron (TDZ, *N*-phenyl-*N*′-l,2,3-thidiazol-5-ylurea) is a substituted phenylurea compound which was synthesized for defoliation of cotton and shows strong CK-like activity in a number of bioassays [24]. TDZ has a strong, positive effect on *Phalaenopsis* micropropagation. TDZ possesses a unique property of mimicking both CK and auxin influences on the morphogenesis of cultured explants. Compared to the amino purine CKs, TDZ effectively promotes callus formation, and induces the higher axillary shoot proliferation and shoot organogenesis of many recalcitrant orchid species at relative lower concentrations (from 0.45 to 4.52 µM) [25,26,27,28]. This may be attributed to a variety of factors, such as an increase in biosynthesis of endogenous CKs and a decrease in cytokinin oxidase activity, or the release of biologically active CKs from nonactive storage forms [29,30,31,32,33].

Although CK treatment is effective in shoot proliferation in *Phalaenopsis* micropropagation, no information is available on the endogenous CKs profiles of cultured *Phalaenopsis* explants. To gain a better understanding of CKs’ role and regulation during the in vitro multiplication of *Phalaenopsis*, we first characterized the key genes participating in the metabolic pathways of CKs, including *IPTs*, *LOGs,* and *CKXs* from the *P. aphrodite* genome database and analyzed the expression profiles of the CKs metabolic genes in various organs and micropropagated shoots during in vitro culture. In addition, we determined endogenous CK-ribosides levels on the TDZ-treated *Phalaenopsis* explants. To further verify and confirm the *PaIPT*, *PaLOG,* and *PaCKX* genes’ functions, we used agroinfiltrated tobacco leaves to overproduce their proteins, and measured the changes of endogenous CK-ribosides levels in tobacco leaves.

## 2. Results

### 2.1. Identification of IPT, LOG, and CKX Proteins in Phalaenopsis

A BLASTP search of the *P. aphrodite* genome database (Genbank accession number: PRJNA383284) was performed using *Arabidopsis* CKs metabolic proteins as a query because most *Arabidopsis* members have been identified, and their functions and tissue specificity are being clarified. In this study, we selected the best match CKs metabolic genes according to their homologous genes in the *A. thaliana* genome. After removing several truncated sequences by manual inspection, a total of six putative PaIPT proteins (Appendix A), six putative PaLOG proteins (Appendix A), and six putative PaCKX proteins (Appendix A) were identified from the *P. aphrodite* genome. Among the six *PaIPT* genes, they contained one or two IPP transferase domains (PF01715), and they distributed on four chromosomes (L04, L08, L10, and L13) and one scaffold (Sca 234). The deduced PaIPT proteins possessed zero to three predicted glycosylation sites (Figure 1A). Among the six *PaLOG* genes, they all contained lysine decarboxylases domains (PF03641), and they distributed on four chromosomes (L03, L05, L06, and L15) and two scaffolds (Sca192 and Sca468). The full lengths of PaLOG proteins ranged from 192 to 244 amino acid residues. None of the proteins contained signal peptide sequences. The deduced PaLOG proteins possessed one predicted glycosylation site (Figure 1B). Among the six *PaCKX* genes, they all contained the CK-binding (PF09265) and FAD-binding (PF01565) domains, and they distributed on three chromosomes (L01, L10, and L15) and two scaffolds (Sca286 and Sca318). The full lengths of PaCKX proteins ranged from 496 to 544 amino acid residues. The majority of PaCKX proteins contained signal peptide sequences with zero to four predicted glycosylation sites (Figure 1C).

### 2.2. Expression of IPT, LOG, and CKX Genes in Different Organs

To understand an initial picture of expressions of *PaIPT*, *PaLOG,* and *PaCKX* genes in different organs during the development, the transcript levels of these genes in leaves, roots, shoots, flower stalks, and flower buds were analyzed by RT-qPCR. Among four *ATP*/*ADP-IPT* genes (*PaIPT1*, *PaIPT3*, *PaIPT4,* and *PaIPT6*) in *Phalaenopsis* genome, *PaIPT1* and *PaIPT3* were highly expressed in flower buds, and also with very high levels in shoots and roots (Figure 2A). Among two *tRNA-IPT* genes, the expression of *PaIPT2* was higher than *PaIPT5* in all organs examined. Among the six *PaLOG* genes, *PaLOG2* was expressed in the greatest abundance in shoots, roots, and flower buds, and *PaLOG4* was predominantly expressed in shoots. *PaLOG5* was highly expressed in shoots, flower stalks, and flower buds (Figure 2B). Among the six *PaCKX* genes, *PaCKX4* was expressed in the greatest abundance in roots, shoots, and flower buds, but it was barely detected in flower stalks (Figure 2C). *PaCKX1* was similar to the expression pattern of *PaCKX4*, but with less abundance. The transcripts of *PaCKX2* and *PaCKX5* were detected in all organs examined with the higher abundance in roots and flower stalks. *PaCKX3* was predominantly expressed in flower buds, but it could not be detected in shoots. Consequently, we selected these CKs metabolic genes with relatively high abundant transcripts in shoots, i.e., three *PaIPTs* (*PaIPT1*, *PaIPT2,* and *PaIPT3*), three *PaLOGs* (*PaLOG2*, *PaLOG4,* and *PaLOG5*), and four *PaCKXs* (*PaCKX1*, *PaCKX2*, *PaCKX4,* and *PaCKX5*) for the further study on their expression profiles during micropropagation by TDZ treatments.

### 2.3. Effect of TDZ Treatments on Shoot Multiplication In Vitro

The shoot proliferation was significantly improved with the increased TDZ concentrations in the stem node culture (Figure 3). In the control without TDZ supplement, a strong apical dominance without the formation of axillary shoot bud was observed (Figure 3A). In TDZ treatments, the basal part of the original shoot explant became swollen with the formation of a number of axillary and adventitious shoot buds (Figure 3B). More adventitious shoot buds were recorded in the 4.54 µM TDZ treatment as compared to the 0.45 µM TDZ treatment (Figure 3C).

### 2.4. Expression of IPT, LOG, and CKX Genes in Micropropagated Shoots

The expression patterns of *PaIPT*, *PaLOG,* and *PaCKX* genes were organ-specific (Figure 2), and thus we selected the CKs metabolic genes with high transcript abundance in micropropagated shoots and demonstrated their expression profiles in Figure 4. TDZ treatments induced the expressions of specific *PaIPT*, *PaLOG,* and *PaCKX* genes in micropropagated shoots. Their expressions increased more rapidly and significantly in 4.54 µM TDZ treatment as compared to those in 0.45 µM TDZ treatment and the control. Among three *PaIPT* genes investigated, the expression of *ATP*/*ADP-IPT* genes, i.e., *PaIPT1* and *PaIPT3,* increased rapidly after 10 days of culture, especially in 4.54 µM TDZ treatment, and remained at relatively high levels until 30 days of culture (~200- and 20-fold), respectively, while the transcript abundance of a *tRNA-IPT* gene, i.e., *PaIPT2,* was much lower (~5-fold) as compared to *PaIPT1* and *PaIPT3* (Figure 4A). Among three *PaLOG* genes investigated, *PaLOG2* was the most abundant and sensitive to TDZ treatments (~80-fold), while the other *PaLOG* genes, i.e., *PaLOG4* and *PaLOG5*, were also induced by TDZ treatments, but the expression levels were lower as compared to *PaLOG2* (~10- to 15-fold) (Figure 4B). Among four *PaCKX* genes investigated, *PaCKX4* was expressed in the greatest abundance in micropropagated shoots (~50-fold), while the other *PaLOG* genes, i.e., *PaCKX1*, *PaCKX2,* and *PaCKX5*, were also induced by TDZ treatments, but the expression levels were lower as compared to *PaCKX4* (~10- to 20-fold) (Figure 4C).

### 2.5. Effect of TDZ Treatments on Endogenous Cytokinin-Ribosides Levels

In micropropagated shoots, TDZ treatments led to an overall increase in CK-ribosides levels (Figure 5). TDZ at 4.54 µM rapidly promoted the accumulation of *trans*-zeatin riboside (*t*ZR) and isopentenyladenine riboside (iPR) by 10 days of culture. Maximal *t*ZR and iPR contents (856.92 and 274.82 pmol/g FW, respectively) were measured at 30 days of culture (Figure 5A,C). The increase in CK-ribosides contents was smoother in 0.45 µM TDZ treatment as compared to that in 4.54 µM TDZ treatment during 30 days of culture. The contents of *cis*-zeatin riboside (*c*ZR) remained at a relatively low level (less than 0.73 pmol/g FW) in all treatments (Figure 5B).

### 2.6. Transient Overexpression of PaIPT1, PaLOG2, and PaCKX4 in Tobacco Leaves

To verify that *PaIPT*, *PaLOG,* and *PaCKX* genes have IPT, LOG, and CKX enzyme activities, we selected the highly expressed *PaIPT1* (*ATP*/*ADP-IPT*), *PaLOG2,* and *PaCKX4* genes in micropropagated shoots and used agroinfiltrated tobacco leaves to overproduce their proteins. CK-ribosides levels in agroinfiltrated tobacco leaves were increased approximately 5- and 4-fold with overexpression of *35S::PaIPT1* (Figure 6A) and *35S::PaLOG2*, respectively (Figure 6B), as compared to those in the controls (transiently overexpressed *35S::GFP* tobacco leaves). In agroinfiltrated tobacco leaves, *t*ZR and iPR increased remarkably, while *c*ZR showed no remarkable difference compared to the control. On the other hand, overexpression of *35S::PaCKX4* in tobacco leaves resulted in a dramatic reduction (approximately six-fold) of CK-ribosides levels (Figure 6C). In addition, the expressed protein was detected with the aid of an anti-His tag mouse antibody. Western blot analysis confirmed that PaCKX4, PaLOG2, and PaIPT1 proteins were successfully expressed in these agroinfiltrated leaves (Appendix A).

## 3. Discussion

CKs are central growth regulators of cell division and development in plants, including apical dominance, branching, photosynthesis, floral transition, seed germination, and organ senescence [4,5]. *Phalaenopsis* has become one of the most important potted plants in the horticultural market, and the use of CKs has been shown to promote multiple floral spikes in potted plant production and to stimulate lateral shoot growth in micropropagation [1]. In spite of its importance, knowledge about how CKs metabolic genes modulate growth and development in orchids is still limited. Identification and analysis of key CKs metabolic genes in *Phalaenopsis* should provide the foundation for understanding of how CK biosynthesis integrates into the developmental regulation in *Phalaenopsis* micropropagation.

In the present study, we identified six *IPT* genes from the *P. aphrodite* genome database. Among the ATP/ADP-type *PaIPT* genes, *PaIPT1* and *PaIPT3* were highly expressed in the reproductive organs, e.g., flower buds, as well as vegetative organs, e.g., shoots and roots (Figure 2A). These organs contain meristematic tissues with metabolically active dividing cells. High abundance of *PaIPT1* and *PaIPT3* transcripts may reflect a key role of CKs during the development of flower buds, shoot, and root tips. Expressions of two other ATP/ADP-type *PaIPT* genes, i.e., *PaIPT4* and *PaIPT6*, were barely detected. Among the tRNA-type *PaIPT* genes identified, expression of *PaIPT2* was higher than *PaIPT5* in all organs examined (Figure 2A). Several CKs metabolic enzymes are encoded by members of gene families, and their isoforms demonstrate various spatial and temporal expression patterns and functional redundancy [16,18,34]. Our data suggest that *Phalaenopsis* requires different *IPT* genes for the biosynthesis of CKs in different organs during development.

In the tissue culture experiment, the expression of *PaIPT1* increased rapidly in 4.54 µM TDZ treatment (~200-fold) (Figure 4A). As a member of *ATP*/*ADP-IPT* genes, *PaIPT1* plays a key role in the regulation of CK-ribosides levels in plant growth and developmental processes [35,36]. The results of *35S::PaIPT1* agroinfiltration assay indicated that *PaIPT1* can promote CK-ribosides levels in tobacco leaves (Figure 6A). Thus, our data suggest that *PaIPT1* is responsible for CK biosynthesis in *Phalaenopsis* shoots. *PaIPT2* belongs to *tRNA-IPT* gene, which is responsible for the synthesis of *c*Z-type CKs. *c*ZR shows a mild CK activity, and *tRNA-IPT* genes are usually unaffected by the application of exogenous plant growth regulators [16,37]. This is in good agreement with our results, where the expression of *PaIPT2* was slightly induced by TDZ (Figure 4A), and there was only a small amount of *c*ZR detected in micropropagated shoots (Figure 5B). Furthermore, application of exogenous CKs represses the expression of *IPT* genes in *Arabidopsis* [16], while this does not appear to be the case in rice [21] and in our results of *Phalaenopsis*. The variation in *IPT* genes’ regulation by CK treatment may reflect differences between plant species in their circuitry of the feedback regulation [21].

LOG has phosphoribohydrolase activity, which directly converts inactive CK nucleotides, e.g., iPRMP and *t*ZRMP, into biologically active free-base forms, e.g., iP and *t*Z [38]. Among the *PaLOG* genes, *PaLOG2* transcript was the most abundant in shoots, roots, and flower buds (Figure 2B). Meanwhile, *PaLOG4* was characterized by predominant expression in shoots. In addition, *PaLOG5* was primarily expressed in flower buds, as well as stalks and shoots. The expression of *LOG* genes with distinct patterns in various tissues and organs has been described in rice [38], *Arabidopsis* [18], and grape [39]. In *Phalaenopsis*, the differentiated accumulation of *PaLOG* transcripts in developing flower buds, shoot, and root tips, especially *PaLOG2* and *PaLOG4*, suggests a dominant CK-activating mechanism through the LOG-dependent pathway.

In the tissue culture experiment, *PaLOG2* transcript was the most abundant (~80-fold) (Figure 3B). Its expression increased sharply by 10 days of 4.54 µM TDZ treatment and was highly expressed during the shoot multiplication, implying an important role of TDZ treatment in regulating CK activity during the shoot multiplication. The results of *35S::PaLOG2* agroinfiltration assay indicated that *PaLOG2* can enhance CKs levels in tobacco leaves (Figure 6B), suggesting a key regulatory function for CKs biosynthesis. Overall, the increase in *t*ZR and iPR levels in micropropagated shoots with the significantly upregulated *PaIPT1* and *PaLOG2* implied that CKs metabolism could be altered by TDZ treatments.

Endogenous CK concentrations need to be dynamically adjusted to specific cell types to optimize plant growth and development. CKX is responsible for the irreversible CK degradation that plays a key role in regulating endogenous CKs in plant tissues [12,40]. Similar to other CKs metabolic genes described above, i.e., *IPT* and *LOG* genes, the expression of specific *IPT* genes in *Phalaenopsis* plants at certain developmental stages and organs seems to be highly regulated (Figure 2C). In the tissue culture experiment, TDZ treatments induced the expressions of *PaCKX1*, *PaCKX2*, *PaCKX4,* and *PaCKX5* (Figure 4C). Among the *PaCKX* genes examined, *PaCKX4* was highly expressed in micropropagated shoots (~50-fold), suggesting a major regulator of CK degradation in the shoot. The results of *35S::PaCKX4* agroinfiltration assay suggested that *PaCKX4* can dramatically reduce CKs levels in tobacco leaves (Figure 6C). Thus, our data demonstrate a possible functionality of *PaCKX4* in CK degradation. Our results are consistent with previous studies, in which exogenous application of CKs induced increases in CKX activity [41,42]. In *Dendrobium*, antisense transgenic plants of *DSCKX1* showed rapid proliferation of shoots and inhibition of root growth, as compared to a higher endogenous CKs content than wild-type plants [43]. It is notable that although the expressions of *PaCKXs* increased in micropropagated shoots (Figure 4C), the contents of CK-ribosides are significantly elevated in TDZ treatments. This may be attributed to an inhibitory effect of TDZ on CK oxidase activity [31].

*Phalaenopsis* has a naturally strong monopodial growth habit, and the growth of axillary meristems is inhibited by the primary shoot apex, a phenomenon commonly known as apical dominance [1]. The axillary shoot proliferation in vitro usually requires a higher concentration range of amino purine CKs. In this study, the exogenous application of TDZ significantly promoted shoot multiplication in *Phalaenopsis* (Figure 3). In the control without TDZ, the outgrowth of axillary bud was strongly inhibited (Figure 3A). Similar results were observed in previous studies with orchid micropropagation [26,44], where TDZ promoted the outgrowth of axillary shoots (Figure 3B) as well as the formation of adventitious shoots (Figure 3C). In *Phalaenopsis*, the basal part of shoot explant became swollen (Figure 3A), suggesting the initiation of adventitious shoot meristems. In *Arabidopsis*, the formation of adventitious shoots required de novo activation of *WUSHEL* (*WUS*), a key regulator of shoot meristem activity during the initiation of adventitious shoot meristems. This activation is mediated by direct binding of type-B *Arabidopsis* response regulator proteins (type-B ARRs) to the *WUS* promoter region [45]. Although TDZ is a synthetically derived urea, it has considerable activity with CK receptors, i.e., CRE1/AHK4 and AHK3, indicating a direct action of TDZ to trigger CK signaling pathways [46,47]. These reports may suggest a direct connection between TDZ treatment, CK signaling, and the regulation of adventitious shoot development.

In addition to the direct action of TDZ in CKs signaling, the elevation of endogenous CK-ribosides levels stimulated by TDZ should be involved in regulation of shoot multiplication. Previous studies have shown that TDZ application promoted induction of synthesis or accumulation of endogenous CKs in plant tissues [29,48]. Our results showed that TDZ treatment significantly increased the contents of *t*ZR and iPR during shoot multiplication (Figure 5). On the contrary, the contents of *c*ZR did not elevate significantly. In 4.54 µM TDZ treatment, after 10 days of culture, *t*ZR levels and iPR levels increased sharply at the time of launching axillary and adventitious shoot buds.

## 4. Materials and Methods

### 4.1. Identification of Phalaenopsis IPT, LOG, and CKX Genes

To identify IPTs, LOGs, and CKXs from *P. aphrodite* genome, a BLASTP (protein basic local alignment search tool) search was performed with the amino acid sequences of CK metabolic proteins in *Arabidopsis thaliana*, i.e., AtIPTs, AtLOGs, and AtCKXs, (http://www.Arabidopsis.org/, accessed on 20 November 2020) as queries against *P. aphrodite* genome databases [21] by the default search parameters. The predicted amino acid sequences as queries were checked by blasting *Arabidopsis* genome databases using the default search parameters. The predicted protein sequences were verified with the Pfam (http://pfam.xfam.org/, accessed on 20 November 2020) and the Simple Modular Architecture Research Tool database (SMART; http://smart.embl-heidelberg.de/, accessed on 20 November 2020) with an E-value cut-off of 10^−10^. Sequences without the related domain were discarded from further analyses. Candidate genes were filtered and identified using the Conserved Domain Search Service (CD-Search) [49]. Their glycosylation site and subcellular localization were predicted by NetNGly (https://services.healthtech.dtu.dk/service.php?NetNGlyc-1.0, accessed on 20 November 2020) and TargetP 2.0 (http://www.cbs.dtu.dk/services/TargetP/, accessed on 20 November 2020). *PaIPT* genes could be mapped on the linkage groups and were renamed from *PaIPT1* to *PaIPT6*. The same order was applied to *PaLOGs* and *PaCKXs*.

### 4.2. Plant Materials and Tissue Culture Experiments

In the tissue culture experiment, blooming plants of *P. aphrodite* were used as the donor plants. At the time of anthesis of the first flower, the second to fourth nodes, including their lateral buds (counted from the base of flower stalks), were collected as source materials for tissue culture. Nodal segments (3 cm) were immersed in 1% sodium hypochlorite solution with 0.1% Tween 20 (Sigma-Aldrich, St.Louis, MO, USA) for 30 min. After three rinses with sterile distilled water, the nodal segments were placed on Murashige and Skoog medium [50], which contained half-strength macro-elements with full-strength microelements and supplemented with 10% fresh coconut water and 20 g.L^−1^ sucrose and solidified with 3 g.L^−1^ Gelrite (Sigma-Aldrich, St.Louis, MO, USA). The pH value of medium was adjusted to 5.7 prior to autoclaving at 121 °C and 1.2 kg·cm^–2^ for 20 min. Each nodal segment was inoculated into 10 mL medium in a 24 × 100 mm test tube and incubated at 27 ± 1 °C under a 12 h photoperiod provided by fluorescent lamps at 35 μmol m^−2^ s^−1^. After 2 months of culture, developing shoots (shoot length of 12 mm) from the lateral buds of nodal segments were harvested and used as explants for subsequent studies. To examine the effect of TDZ on the shoot multiplication, the changes of endogenous CK levels, and the expression of CKs metabolic genes, explants were inoculated on the half-strength Murashige and Skoog medium supplemented with 0.45 and 4.54 µM TDZ for 10, 20, and 30 days, respectively. The medium lacking TDZ supplement served as control.

### 4.3. Quantitative RT-PCR (RT-qPCR)

For measuring the expression profiles of CKs metabolic genes in various organs, mature plants of *P. aphrodite* in 12 cm pots filled with sphagnum moss were grown in the growth chamber at 60 μmol m^–2^ s^–1^ with the 12 h light (22 °C)/12 h dark (20 °C) cycles with regular irrigation and fertilization in the Institute of Medicinal Plant Development, Chinese Academy of Medical Sciences and Peking Union Medical College, Beijing. The first mature leaf (from the top) with a length of 20 cm, the active growing root with a length of 6 cm, the shoot tip containing a few leaf primordia with a length of 1 cm (after removing leaves), the first internode of flower stalk (below the first flower bud at the time of anthesis), and the developing flower bud with a length 5 mm were collected and immediately frozen in liquid nitrogen, then stored at −80 °C until analyses. For measuring expression profiles of CKs metabolic genes in micropropagated shoots, shoot explants containing a few leaf primordia with a length of 1 cm at different days of culture were collected and frozen in liquid nitrogen as described above. Total RNA was extracted from these samples using the TRIZOL^TM^ reagent (Invitrogen, Waltham, MA, USA). The purified RNA was then treated with RQ1 DNase (Invitrogen, Waltham, MA, USA) to remove genomic DNA contamination. The cDNA was synthesized using the PrimeScript RT reagent Kit (TaKaRa Bio, Shiga, Japan) according to the manufacturer’s instructions. For real-time PCR, a master mix for each PCR run was prepared with SYBR Premix Ex Taq II (TaKaRaBio) involving 7.5 µL of SYBR Premix Ex Taq II, 1.5 µL cDNA, and 0.5 µL primers, and water was added to 15 µL. Each sample was analyzed in three biological replicates with three technical replicates by using the Light Cycler 480 II Real-Time PCR System (Roche, Basel, Switzerland) with its relative quantification program. The amplification program was used: initial denaturation at 95 °C for 30 s, then 40 cycles of 95 °C for 5 s, followed by 60 °C for 30 s. The *PaUbiquitin* gene was used as an internal quantification standard. The 2^−ΔΔCt^ method was used for evaluating gene expression. Fold changes were normalized to their corresponding controls (the respective non-treated micropropagated shoot at 0 day, 10 days, 20 days, or 30 days after culture). The primers used for real-time PCR were designed by using Premier 5.0 software and were provided in Appendix A.

### 4.4. Transient Overexpression of PaIPT, PaLOG, and PaCKX Proteins in Tobacco Leaves

For overexpressing PaIPT, PaLOG, and PaCKX proteins in plant cells, cDNA fragments of highly expressed *PaIPT1* (*ATP*/*ADP-IPT*), *PaLOG2,* and *PaCKX4* genes were amplified by RT-PCR, then digested by PstI and BamHI restriction endonucleases, and introduced into the pEpyon32H binary vector to produce the p32H-IPT plasmid under control of a 2× 35S CaMV promoter. After sequencing confirmation, the plasmids were transformed into *Agrobacterium* strain C58C1. Agroinfiltration was performed according to Cheng et al. [51] on leaves of 4-week-old *Nicotiana benthamiana*. The transiently overexpressed *35S::GFP* tobacco leaves were used as their controls. After transient expression for 3 days under a 16/8 h photoperiod at 25 ± 2 °C, content of endogenous CKs was measured in agroinfiltrated tobacco leaves as described above.

Furthermore, the Western blot assay was used to confirm their protein expressions. A total of 100 mg of agroinfiltrated leaves was mixed with protein extraction buffer, vortexed for 3 min, centrifuged at 10,000× *g* at 4 °C, and then the supernatant was mixed with sodium dodecyl sulfate (SDS) loading buffer in a 3:2 ratio at 100 °C for 10 min. Total proteins were subjected to SDS–polyacrylamide gel electrophoresis. After being separated, the proteins were transferred to polyvinylidene fluoride membranes and saturated with 5% skimmed milk in TBST buffer for 2 h. The membranes were incubated with 6× His antibody (CWBIO, Nanjing, China) at 4 °C overnight. After being washed three times with TBST, the membranes were incubated with horseradish peroxidase conjugated goat anti-mouse IgG (CWBIO, Nanjing, China) as the secondary antibody for 1 h. According to the instructions of Immobilon Western Chemiluminescent HRP Substrate Kit (EMD Millipore corporation, Billerica, MA, USA), the working HPR substrate was used to cover the membranes, which were incubated for 3 min in the dark until the signal was clearly visible.

### 4.5. Quantification of Endogenous CK-Ribosides

The method for extraction and purification of cytokinins has been described in detail by Bollmark et al. [52]. Micropropagated shoots with the length of 1 cm and the fresh weight of 1 g were harvested from each culture condition, immediately frozen in liquid nitrogen, and stored at −80 °C until analyses. Samples were ground in an ice-cooled mortar in the extraction solution (80% methanol, 2% glacial acetic acid). The extract was incubated at 4 °C for 24 h under darkness, then centrifuged at 4000 rpm for 15 min at the same temperature. The supernatant was passed through a C18 cartridge (Waters Corp., Millford, MA, USA), prewashed with 10 mL 100% (*w*/*v*) and 5 mL 80% (*v*/*v*) methanol, respectively. The hormone fractions eluted with 10 mL 100% (*v*/*v*) methanol and 10 mL ether from the columns were dried under N_2_ and dissolved in 2 mL phosphate buffer saline (PBS) containing 0.1% (*v*/*v*) Tween 20 and 0.1% (*w*/*v*) gelatin (pH 7.5) for ELISA analysis. Quantification of CK ribosides was estimated by indirect ELISA method according to Eberle et al. [53] and Yang et al. [54] using plant hormones immunoassay detection kits (Agrisera, Vännäs, Sweden).

### 4.6. Experimental Design and Data Analysis

Experiments of shoot multiplication were performed in a randomized design and repeated three times. Each treatment had 15 replicates and each replicate consisted of three explants per culture vessel. The average number of shoot and adventitious shoot per explant was recorded after 30 days of culture. In order to examine whether shoot multiplication, hormone concentration, and gene expression were significantly different between the treatments, the quantitative data were analyzed by unpaired Student’s *t*-test, and significant difference was set at *p* < 0.05 and *p* < 0.01 or by one-way analysis of variance (ANOVA), and the post hoc test was performed with Tukey’s honest significant difference (HSD) test. The significance level was set at *p* < 0.05.

## 5. Conclusions

Our study elaborates how TDZ triggers in vitro shoot multiplication of *Phalaenopsis* through inducing de novo biosynthesis of CKs, primarily in the elevation of *t*Z-type and iP-type CK levels. The most relevant CKs biosynthetic genes involved in this process are *PaIPT1* and *PaLOG2*. Interestingly, TDZ induces *PaCKX4* expression, a major CK oxidation gene, but TDZ also has an inhibitory effect on CK oxidase. Our results present the first insight into the previously uncharacterized CKs metabolic genes in *Phalaenopsis* genome and provide a better theoretical reference for applying TDZ in practice, e.g., in vitro experiments in *Phalaenopsis* related to developmental studies and the optimized duration of TDZ treatments in mass production.

## Figures and Tables

**Figure 1 ijms-23-03755-f001:**
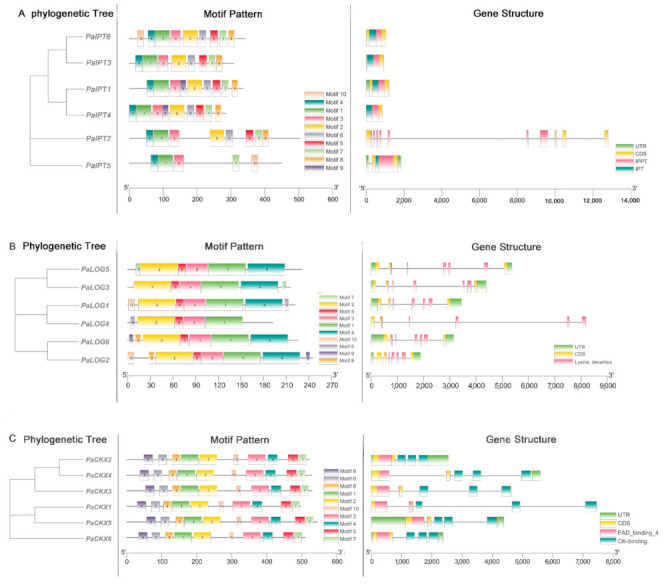
The gene structure, motifs distribution, and phylogenetic relationship of *PaIPTs* (**A**), *PaLOGs* (**B**), and *PaCKXs* (**C**). For phylogenetic tree, the analyses were constructed based on protein sequences using MEGA-X software. For conserved motif analyses, the motifs are presented in different colored boxes with numbers 1–10. The length of gene can be estimated using the scale at the bottom. For gene structure analyses, green boxes indicate untranslated 5′- and 3′-regions, yellow boxes indicate exons, and black lines indicate introns. The IPPT, Lysine_decarbox, and FAD_binding_4 domain are highlighted by red boxes with IPT and CK-binding domain in dark green.

**Figure 2 ijms-23-03755-f002:**
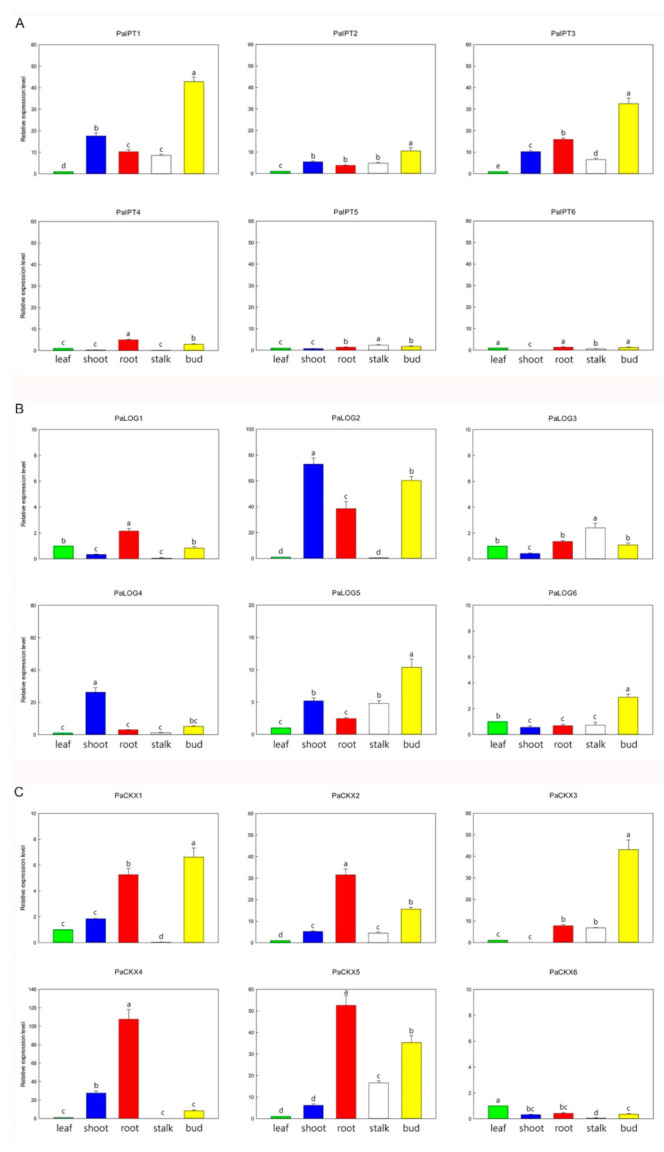
Relative expression levels of *PaIPTs* (**A**), *PaLOGs* (**B**), and *PaCKXs* (**C**) in various organs, including leaf, shoot, root, flower stalk (stalk), and flower bud (bud). Data are the mean of three biological replicates and error bars represent ± standard deviations. Data were analyzed using one-way ANOVA with Tukey’s honest significant difference (HSD) test, and significant differences are indicated by different letters at *p* < 0.05 level.

**Figure 3 ijms-23-03755-f003:**
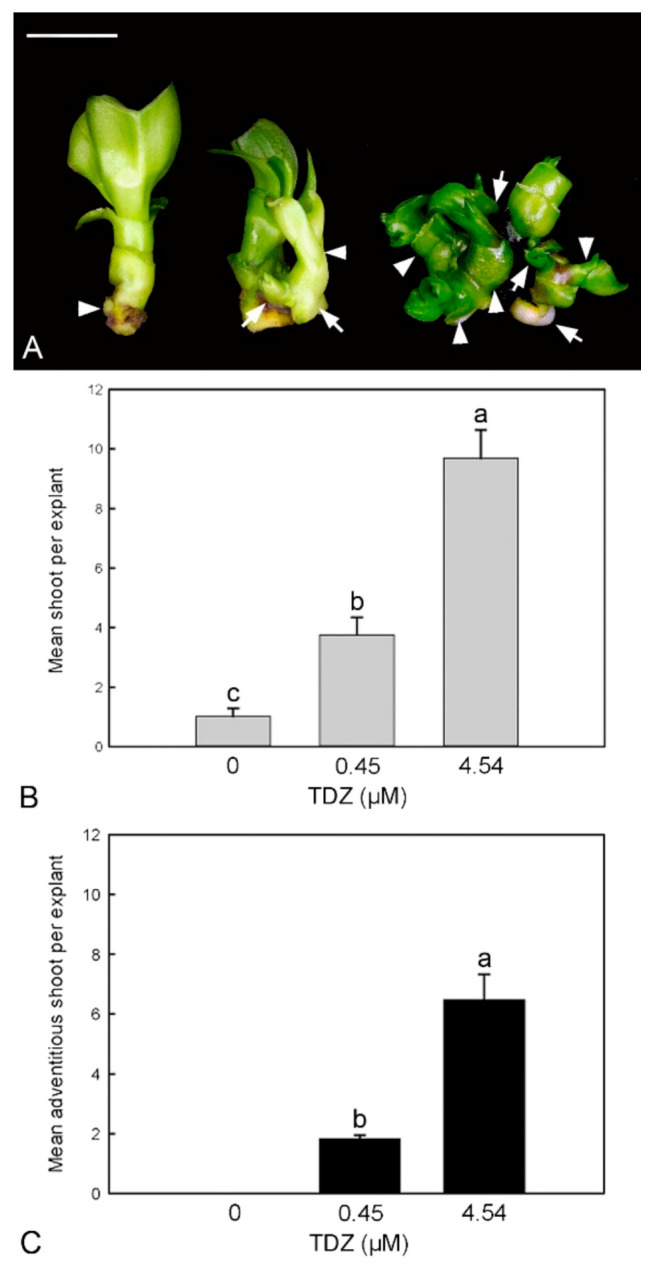
The effect of TDZ treatments on shoot multiplication in vitro. (**A**) Micropropagated shoots after 30 days of inoculation by TDZ treatment. In the control (**left**), the micropropagated shoot showed strong apical dominance without shoot multiplication. The axillary shoot bud (arrowhead) did not become enlarged. In 0.45 µM TDZ treatment (**central**), one axillary shoot bud (arrowhead) developed from the basal of the original micropropagated shoot, then it further produced two adventitious shoot buds (arrows). In 4.54 µM TDZ treatment (**right**), the formation of a number of axillary shoot buds (arrowheads) and adventitious shoot buds (arrows) from the original micropropagated shoot (right). Bar = 1 cm. (**B**) Effect of TDZ treatments on the mean number of shoot bud per explant after 30 days of culture. (**C**) Effect of TDZ treatments on the mean number of adventitious shoot bud per explant after 30 days of culture. Data were analyzed using one-way ANOVA with Tukey’s honest significant difference (HSD) test, and significant differences are indicated by different letters at *p* < 0.05 level.

**Figure 4 ijms-23-03755-f004:**
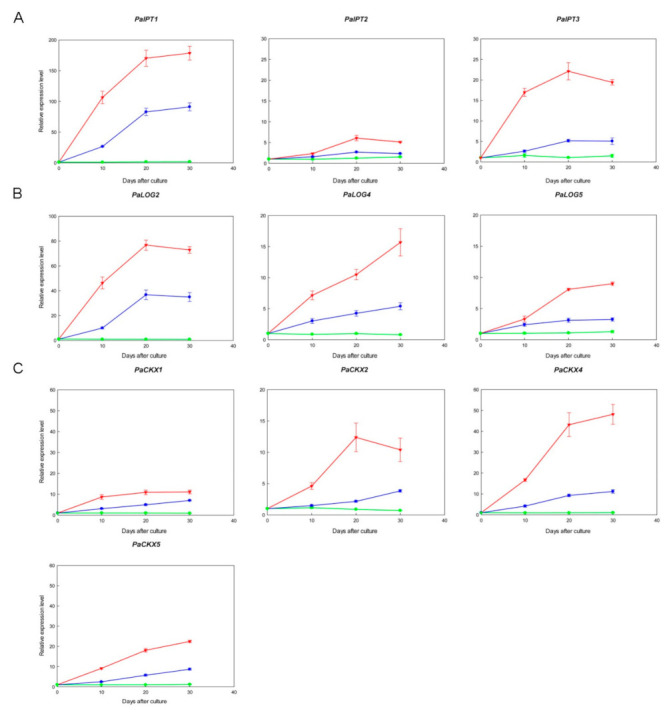
Transcript levels of cytokinins metabolic genes in micropropagated shoots inoculated at 0 (control, green), 0.45 (blue), or 4.54 (red) µM TDZ for 30 days. (**A**) *PaIPTs*; (**B**) *PaLOGs*; (**C**) *PaCKXs*. Fold changes (relative expression levels) are normalized to the corresponding controls (the respective non-treated micropropagated shoot at 0 day, 10 days, 20 days, or 30 days after culture). Data are the mean of three biological replicates and error bars represent ± standard deviations.

**Figure 5 ijms-23-03755-f005:**
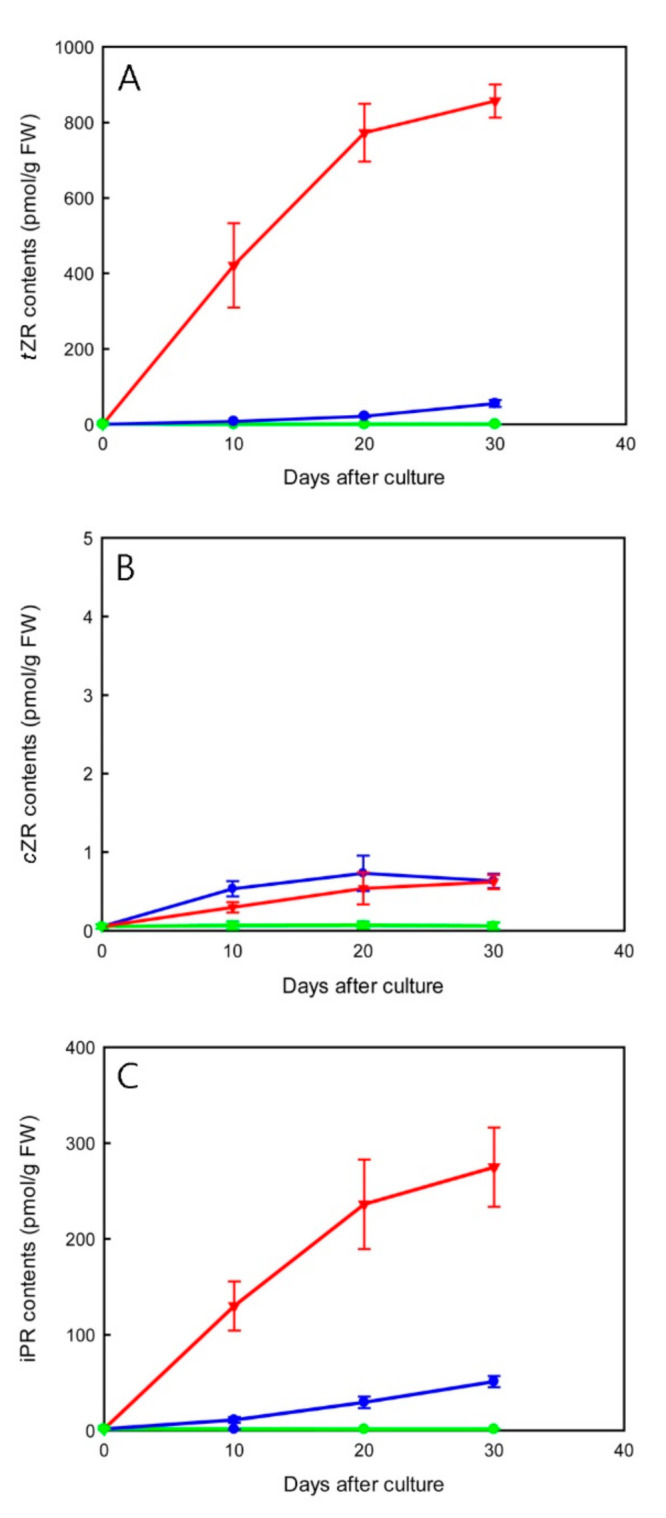
The effect of TDZ treatments on changes of total contents (pmol/g FW) of CK ribosides (**A**) *t*ZR, (**B**) *c*ZR, and (**C**) iPR in micropropagated shoots. Data are the mean of three biological replicates and error bars represent ± standard deviations.

**Figure 6 ijms-23-03755-f006:**
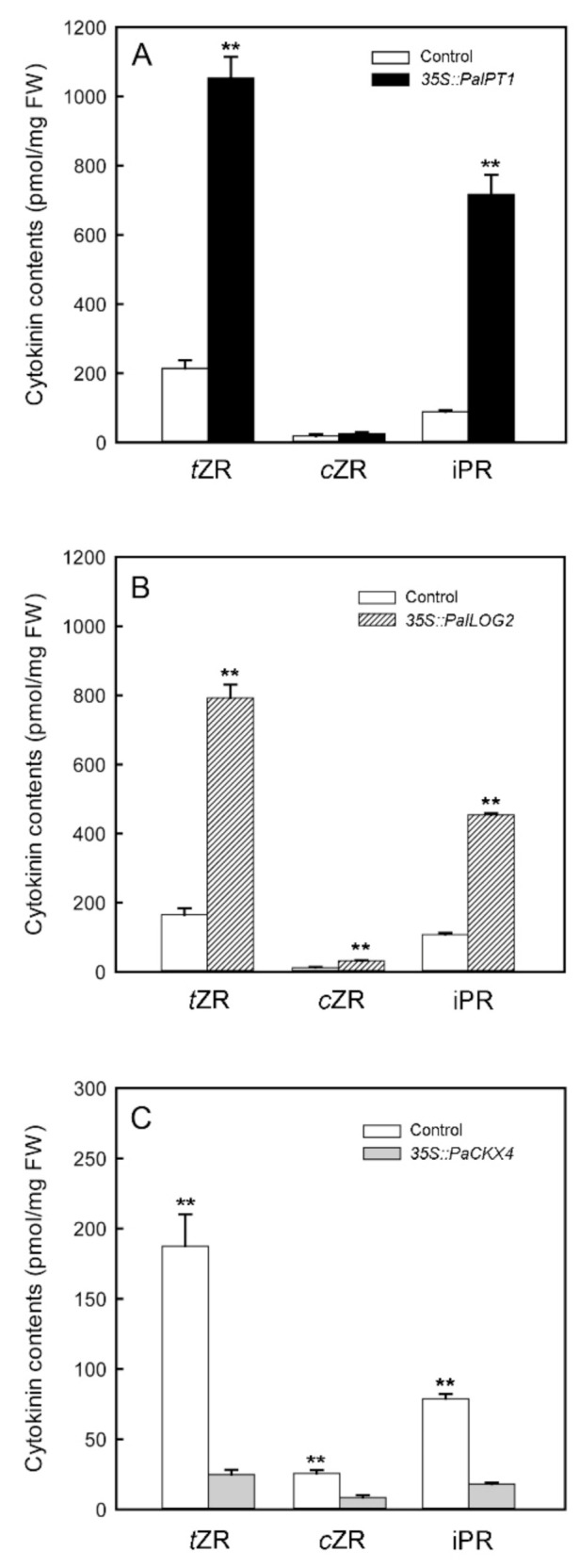
Levels of cytokinins in the *PaIPT1* (**A**), *PaLOG2* (**B**), and *PaCKX4* (**C**) transiently overexpressed tobacco leaves. The transiently overexpressed *35S::GFP* tobacco leaves were used as their controls. Data are the mean of three biological replicates and error bars represent ± standard deviations. Data were analyzed by unpaired Student’s *t*-test, and significant differences were set at *p* < 0.01 (**).

## Data Availability

Not applicable.

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
