# Peer review of "Profiles of Cytokinins Metabolic Genes and Endogenous Cytokinins Dynamics during Shoot Multiplication In Vitro of Phalaenopsis"

_ijms, 2022, doi:10.3390/ijms23073755_

Round 1
Reviewer 1 Report
In this article, the authors identified Phalaenopsis genes involved in the biosynthesis of cytokinin. They evaluate the expression profile of these genes during the development of different organs in Phalaenopisis, and the effect of an artificial plant growth regulator – thidiazuron— on the expression of those genes and the accumulation of cytokinin during Phalaenopsis micropropagation. Using a transient overexpressing system in tobacco leaf tissue, they demonstrated the importance of those genes for cytokinin accumulation. This paper provides the reader with novel information about the molecular mechanisms underlying cytokinin metabolism during Phalaenopsis micropropagation.
Specific comments about the paper are described below:
The abstract is not well-structured, it does not summarize well the results, it does not highlight the novelty of the paper. For example, the abstract does not mention clearly that no CKs metabolic genes have been characterized previously in Phalaenopsis, and it mentions random results from the paper. The abstract must be fully edited.
The introduction is well-structured, and the information given is relevant, although English language and style need in-depth editing. Specific comments about the introduction are listed below:
_ Line 44 to 50: references must be provided here.
_ Line 52 to 56: this paragraph briefly mentions the uptake and metabolism of exogenous and endogenous CKs. The reader can be provided here with more bibliographical details about the differences in the effect of exogenous and endogenous CKs – this might help to hypothesize on the mechanism of action of TDZ in planta, the latter being presented in the introduction later.
_ Line 78 to 88: The authors mention the use of TDZ for Phalaenopsis micropropagation and the absence of understanding of its mechanism of action on the CK biosynthesis. However, they do not provide with information about the current qualitative and quantitative importance of its use for Phalaenopsis micropropagation. This might be mentioned here to highlight the importance of this research study.
The results section is well-structured. However, there is a lack of information throughout the result section and in the figures that prevent the reader to assess properly the quality of the experimental design or the validity of the results given. Please respond to the specific comments below:
Results, section 2.1:
_ Table 1, 2 and 3 are supporting information that should be presented in the supplemental data and not in the main figures. The authors must specify the differences between the PFAM e-value IPPT and PFAM e-value IPT in the legend.
_ Figure 1 must be explained thoroughly in a detailed legend that is currently missing.
_ line 106 to 109: this sentence must be re-formulated.
_ general comment of the results, part 2.1: I do not understand here why the predicted glycosylation sites specifically are indicated by the authors. There are other important features that have not been considered. How does it help the general message delivered? Importance of glycosylation for these candidates is not discussed further in the Results or Discussion section.
Results, section 2.2:
_ Figure 2: The legend does not provide sufficient information about the figures, and details about the statistical analyses and grouping used in the figures to assess the quality of the information. This must be added here.
Results, section 2.4:
line 206: ‘ the root initiation was greatly suppressed by TDZ-treatments during shoot multiplication’. This is not supported by the data presented and should therefore not be mentioned. An additional figure should be added by the authors to support this information.
Figure 4: ‘Fold changes are normalized to the corresponding controls (the micropropagated shoot before TDZ-treatments at day zero).’ The micropropagated shoot before TDZ-treatments at day 0 is not the right control to use. The right control to use for the TDZ-treated micropropagated shoot at 10 days, 20 days and 30 days after culture should be the respective non-treated micropropagated shoot at 10 days, 20 days and 30 days. In fact, one can assume that the endogenous levels of cytokinin fluctuates between 0 and 30 days of culture, and gene expression of CK-associated metabolic genes can fluctuate too.
Results, section 2.5:
Figure 5: The legend should specify if data is the mean of three technical or biological replicates.
Results, section 2.6:
Western blotting pictures are of very poor quality and cannot be published as it is, and there is not enough information provided in the supplemental data or on the original blot to assess properly the quality of the results provided. Without this control information, Figure 6 is not valid.
Figure 6 does not mention which control has been used. This needs to be specified clearly in the figure and in the result section.
Although the bibliographical references given in the Discussion support the arguments presented and the data seem to be interpreted adequately, the discussion is not well-structured, the conclusions are not comprehensive. The author must also be careful with the interpretation of the data. The effect of over-expression of CKX4, LOG2 and IPT1 can be dramatically different between tobacco leaf tissue and Phalaenopsis micropropagated tissues. Any conclusion resulting from transient expression of genes in another biological system must be interpreted carefully as it is not a sufficient proof for explaining the molecular mechanisms driven by those genes.
I would also suggest the authors to add a figure with a hypothetic model that summarizes and highlight the information provided in the paper, this would summarize the results and discussion adequately for the reader.
Author Response
Specific comments about the paper are described below:
The abstract is not well-structured, it does not summarize well the results, it does not highlight the novelty of the paper. For example, the abstract does not mention clearly that no CKs metabolic genes have been characterized previously in Phalaenopsis, and it mentions random results from the paper. The abstract must be fully edited. The introduction is well-structured, and the information given is relevant, although English language and style need in-depth editing. Specific comments about the introduction are listed below:
We appreciate your comments. We have rewritten the abstract and described the key information orderly from the results (L16-L23).
Line 44 to 50: references must be provided here.
We have added references to the text (L48-L50).
Line 52 to 56: this paragraph briefly mentions the uptake and metabolism of exogenous and endogenous CKs. The reader can be provided here with more bibliographical details about the differences in the effect of exogenous and endogenous CKs – this might help to hypothesize on the mechanism of action of TDZ in planta, the latter being presented in the introduction later.
Thank you for this suggestion. We have added more details about the effect of exogenous and endogenous CKs to the text (L54-L56).
Line 78 to 88: The authors mention the use of TDZ for Phalaenopsis micropropagation and the absence of understanding of its mechanism of action on the CK biosynthesis. However, they do not provide with information about the current qualitative and quantitative importance of its use for Phalaenopsis micropropagation. This might be mentioned here to highlight the importance of this research study.
We have provided the qualitative and quantitative information of TDZ used in micropropagation (L87-L91).
The results section is well-structured. However, there is a lack of information throughout the result section and in the figures that prevent the reader to assess properly the quality of the experimental design, or the validity of the results given. Please respond to the specific comments below:
Results, section 2.1:
Table 1, 2 and 3 are supporting information that should be presented in the supplemental data and not in the main figures. The authors must specify the differences between the PFAM e-value IPPT and PFAM e-value IPT in the legend.
Thanks for your kind advice. We have moved Table 1, 2 and 3 into the supplemental data. The difference between IPT and IPPT has been added to table note. PFAM e-values for having the indicated protein domains. IPT domain means Isopentenyl transferase, and IPPT domain means IPPT transferases in PFAM database (Please see the supplementary data).
Figure 1 must be explained thoroughly in a detailed legend that is currently missing.
We have added a detailed legend for Figure 1 (L132-L137).
line 106 to 109: this sentence must be re-formulated.
Thank you so much for this comment. We have re-formulated this sentence (L111-L112).
General comment of the results, part 2.1: I do not understand here why the predicted glycosylation sites specifically are indicated by the authors. There are other important features that have not been considered. How does it help the general message delivered? Importance of glycosylation for these candidates is not discussed further in the Results or Discussion section.
Thank you so much for this comment. Glycosylation can link saccharides to specific amino acids of the protein that alters physical and chemical properties of the protein. Therefore, it affects the folding, distribution, stability and biological activity of the protein. In the bioinformatic analyses, we showed this feature of three CKs metabolic enzymes. In this study, we do not intend to explore their biochemical properties, and thus we have moved these tables to the supplementary data according to your suggestion (Please see the supplementary data).
Results, section 2.2:
Figure 2: The legend does not provide sufficient information about the figures, and details about the statistical analyses and grouping used in the figures to assess the quality of the information. This must be added here.
We have provided sufficient information and details about statistical analyses of Figure 2 legend (L161-L163).
Results, section 2.4:
line 206: ‘the root initiation was greatly suppressed by TDZ-treatments during shoot multiplication’. This is not supported by the data presented and should therefore not be mentioned. An additional figure should be added by the authors to support this information.
We have removed this sentence according to your suggestion. In Figure 3A, no root formation was observed in these TDZ-treated proliferating shoots.
Figure 4: ‘Fold changes are normalized to the corresponding controls (the micropropagated shoot before TDZ-treatments at day zero).’ The micropropagated shoot before TDZ-treatments at day 0 is not the right control to use. The right control to use for the TDZ-treated micropropagated shoot at 10 days, 20 days and 30 days after culture should be the respective non-treated micropropagated shoot at 10 days, 20 days and 30 days. In fact, one can assume that the endogenous levels of cytokinin fluctuates between 0 and 30 days of culture, and gene expression of CK-associated metabolic genes can fluctuate too.
Thank you so much for this comment. We are sorry that we did not write the control information properly. The controls used for the TDZ-treated micropropagated shoot at 10 days, 20 days and 30 days after culture were the respective non-treated micropropagated shoot at 10 days, 20 days and 30 days (L207, L408-L409).
Results, section 2.5:
Figure 5: The legend should specify if data is the mean of three technical or biological replicates.
Data was the mean of three biological replicates. We have added this information to the legend (L220).
Results, section 2.6:
Western blotting pictures are of very poor quality and cannot be published as it is, and there is not enough information provided in the supplemental data or on the original blot to assess properly the quality of the results provided. Without this control information, Figure 6 is not valid.
We have repeated Western blotting experiments and provided images of better quality with the control using the 35S::GFP. Please see the revised Figure S1.
Figure 6 does not mention which control has been used. This needs to be specified clearly in the figure and in the result section.
Thank you so much for this comment. We have provided the control information (transiently overexpressed 35S::GFP tobacco leaves) and added it in the text of result and M&M sections (L227-L228, L237-L238).
Although the bibliographical references given in the Discussion support the arguments presented and the data seem to be interpreted adequately, the discussion is not well-structured, the conclusions are not comprehensive. The author must also be careful with the interpretation of the data. The effect of over-expression of CKX4, LOG2 and IPT1 can be dramatically different between tobacco leaf tissue and Phalaenopsis micropropagated tissues. Any conclusion resulting from transient expression of genes in another biological system must be interpreted carefully as it is not a sufficient proof for explaining the molecular mechanisms driven by those genes.
Thank you so much for the suggestions. We have revised the text and understated the conclusion resulting from transient expression of PaCKX4, PaLOG2 and PaIPT1 genes in tobacco leaves (L297-L298, L308-L310).
I would also suggest the authors to add a figure with a hypothetic model that summarizes and highlight the information provided in the paper, this would summarize the results and discussion adequately for the reader.
Thank you so much for the suggestions. Our present results may not be sufficient to draw a hypothetic model. Our further study on Phalaenopsis CKs signaling genes may provide sufficient information to draw a hypothetic model.
Reviewer 2 Report
Dear authors;
I think that this study will make an important contribution to the understanding of cytokinins metabolic genes in plants, especially in Phalaenopsis plant. In this context, I believe that the wet-lab part of the study was well planned and the findings were well presented and discussed.
Author Response
I think that this study will make an important contribution to the understanding of cytokinins metabolic genes in plants, especially in Phalaenopsis plant. In this context, I believe that the wet-lab part of the study was well planned and the findings were well presented and discussed.
Thank you so much for the encouragement.
Reviewer 3 Report
This work is an independent and completed research. It cannot be called navatory, but it is a qualitative study, the results of which are not in doubt. However, in my opinion, the authors do not sufficiently cover the background of the issue and the role of cytokinins in general in the growth of cultures in vitro. For example, here are just a couple of works that are very close in meaning to the work of the authors doi.org/10.1007/s11240-019-01693-5 and doi.org/10.1016/j.sajb.2010.02.011, although they are not molecular, but about this also needs to be said. I would also like to draw your attention to the low quality of the figures in 2 and 4, especially to the too small font of the axis labels, which is extremely difficult to read.
Author Response
This work is an independent and completed research. It cannot be called navatory, but it is a qualitative study, the results of which are not in doubt. However, in my opinion, the authors do not sufficiently cover the background of the issue and the role of cytokinins in general in the growth of cultures in vitro. For example, here are just a couple of works that are very close in meaning to the work of the authors doi.org/10.1007/s11240-019-01693-5 and doi.org/10.1016/j.sajb.2010.02.011, although they are not molecular, but about this also needs to be said. I would also like to draw your attention to the low quality of the figures in 2 and 4, especially to the too small font of the axis labels, which is extremely difficult to read.
Thank you so much for the suggestions. We have added the information of these works to our references. We have provided higher resolution images of Figures in 2 and 4.